# MC3G: Model Agnostic Causally Constrained Counterfactual Generation[*]

**Sopam Dasgupta**                                  SOPAM.DASGUPTA@UTDALLAS.EDU
**Sadaf MD Halim**                                  SADAFMD.HALIM@UTDALLAS.EDU
*The University of Texas at Dallas*
**Joaquín Arias**                                   JOAQUIN.ARIAS@URJC.ES
*CETINIA, Universidad Rey Juan Carlos*
**Elmer Salazar**                                   EES101020@UTDALLAS.EDU
**Gopal Gupta**                                     GUPTA@UTDALLAS.EDU
*The University of Texas at Dallas*

**Editors:** Leilani H. Gilpin, Eleonora Giunchiglia, Pascal Hitzler, and Emile van Krieken

## Abstract

Machine learning models increasingly influence decisions in high-stakes settings such as finance, law and hiring, driving the need for transparent, interpretable outcomes. However, while explainable approaches can help understand the decisions being made, they may inadvertently reveal the underlying proprietary algorithm—an undesirable outcome for many practitioners. Consequently, it is crucial to balance meaningful transparency with a form of recourse that clarifies why a decision was made and offers actionable steps following which a favorable outcome can be obtained.

Counterfactual explanations offer a powerful mechanism to address this need by showing how specific input changes lead to a more favorable prediction. We propose Model-Agnostic Causally Constrained Counterfactual Generation (MC3G), a novel framework that tackles limitations in the existing counterfactual methods. First, MC3G is model-agnostic: it approximates any black-box model using an explainable rule-based surrogate model. Second, this surrogate is used to generate counterfactuals that produce a favourable outcome for the original underlying black box model. Third, MC3G refines cost computation by excluding the "effort" associated with feature changes that occur automatically due to causal dependencies. By focusing only on user-initiated changes, MC3G provides a more realistic and fair representation of the effort needed to achieve a favourable outcome.

We show that MC3G delivers more interpretable and actionable counterfactual recommendations compared to existing techniques all while having a lower cost. Our findings highlight MC3G's potential to enhance transparency, accountability, and practical utility in decision-making processes that incorporate machine-learning approaches.

## 1. Introduction

Machine learning models have become indispensable in high-stakes decision-making systems, influencing outcomes in domains such as finance (e.g., loan approvals), law (e.g., risk assessments), and hiring (e.g., candidate screening). Unfortunately, these models often function as black boxes, making it difficult to understand the reasoning behind their

---

[*] This work was supported by the US NSF Grants IIS 1910131 and IIP 1916206, US DoD, grants from industry. We also thank the members of ALPS lab at UT Dallas for the insightful discussions.

decisions. This lack of transparency raises concerns about fairness, accountability, and trust, especially when decisions have significant consequences for individuals. To address these challenges, explainable AI (XAI) techniques aim to provide insights into model behavior. However, a fundamental trade-off exists between explainability and model protection—many organizations rely on proprietary models and are reluctant to disclose internal logic due to competitive, security, or privacy reasons. As a result, there is a growing need for explanations that offer meaningful recourse to affected individuals without exposing model internals. Counterfactual explanations provide a powerful mechanism to achieve this balance. Instead of revealing how a model arrives at a decision, counterfactuals answer the question: "What changes in input features would have led to a different, more favorable outcome?" For instance, in a loan application scenario, a counterfactual explanation might state: "If the applicant's credit score had been 650 instead of 600, the loan would have been approved." Such explanations offer actionable recourse, enabling users to understand what changes are necessary to achieve a desirable outcome.

Existing counterfactual generation methods face key limitations: 1) Not model agnostic: Many approaches require direct access to model parameters, making them impractical for proprietary systems. 2) Ignore causal dependencies: Traditional methods assume features can be altered independently, leading to unrealistic counterfactuals (e.g., arbitrarily increasing credit score without addressing underlying financial history). 3) Inefficient cost computation: Most frameworks treat all feature changes equally, failing to distinguish between user-initiated changes and automatic adjustments due to causal relationships.

To address these limitations, we propose Model-Agnostic Causally Constrained Counterfactual Generation (MC3G), a novel framework that enhances the realism, interpretability, and usability of counterfactual explanations while maintaining model secrecy. Our key contributions are as follows: 1) Model-agnostic framework: Instead of accessing the black-box model directly, MC3G first approximates it using an explainable rule-based model (e.g., FOLD-SE). This allows MC3G to work across different machine learning models without compromising their proprietary nature. 2) Refined cost computation: MC3G separates user-initiated interventions from automatic feature adjustments caused by causal dependencies. This ensures a more accurate and fair representation of effort required to achieve a counterfactual outcome. 3) Improved actionability and interpretability: By enforcing causal constraints, MC3G generates realistic counterfactuals that align with real-world constraints (e.g., increasing credit score realistically through debt reduction). This makes the generated explanations more useful and actionable for individuals seeking recourse.

In the following sections, we formalize the problem setting, present the MC3G methodology and evaluate its performance across multiple datasets. Our findings demonstrate that MC3G outperforms existing counterfactual methods in realism and cost-efficiency, making it a valuable tool for transparency in machine-learning decision systems.

## 2. Background and Related Work

### 2.1. Counterfactual Explanations

Explanations help humans understand decisions and inform actions. Counterfactual explanations (CFE) indicate minimal feature changes that would yield a different outcome, aligning with "what-if" human reasoning. Formally, for a binary classifier $f : X \rightarrow \{0, 1\}$, a counterfactual explanation is defined as an alternative input $\hat{x}$ where the model's prediction

changes: $CF_f(x) = \{\hat{x} \in X | f(x) \neq f(\hat{x})\}$. This set of counterfactual explanations consists of all instances $\hat{x}$ that lead to a different prediction under $f$, compared to the original input $x$. Although widely adopted in XAI, many methods assume feature independence, producing unrealistic interventions. For instance, Boundary Counterfactuals by Wachter et al. (2018) optimize closeness but ignore causality, DiCE by Mothilal et al. (2020) offers diverse solutions yet they do not natively model causal dependencies. MACE by Karimi et al. (2020) finds minimal changes without enforcing causal relationships, and C3G by Dasgupta et al. (2024), while producing causally compliant counterfactuals, is restricted to rule-based models. Thus, a model-agnostic approach that respects causal dependencies and ensures feasible interventions is urgently needed—a gap MC3G addresses.

## 2.2. Causality

Causality explains how changes in one variable affect another. In the Structural Causal Model (SCM) framework by Pearl (2009), interventions—external actions that alter a variable's state—capture true cause-effect relationships rather than mere correlations. When generating counterfactual explanations, it is essential that interventions are feasible and respect these causal dependencies. For instance, in a loan approval scenario, an increase in income may naturally boost credit scores through improved debt repayment; treating these as independent changes can lead to unrealistic recommendations.

## 2.3. C3G: Causally Constrained Counterfactual Generation

Causally Constrained Counterfactual Generation (C3G) by Dasgupta et al. (2024) explicitly models causal dependencies using Answer Set Programming (ASP) to ensure counterfactuals adhere to real-world cause–effect relationships rather than assuming feature independence as seen in methods like Boundary Counterfactuals, DiCE, and MACE. For example, for a loan rejected due to low income and a poor credit score, C3G recognizes that increasing income will naturally improve the credit score, recommending only the direct intervention. However, C3G currently assigns a cost to every feature change—even those automatically induced by causal propagation—potentially overestimating the cost of realistic counterfactuals.

## 2.4. Answer Set Programming (ASP) and s(CASP)

Answer Set Programming (ASP) is a declarative framework for knowledge representation and non-monotonic reasoning, making it well-suited for dynamic environments [Brewka et al. (2011); Baral (2003); Gelfond and Kahl (2014)]. Its goal-directed solver, s(CASP) by Arias et al. (2018), executes programs in a top-down, query-driven manner and employs program completion to transform "if" rules into "if and only if" rules, enabling bidirectional reasoning and the simulation of causal interventions by encoding causal relationships (e.g., $(P \Rightarrow Q) \wedge (\neg P \Rightarrow \neg Q)$). By integrating ASP's reasoning with a refined cost computation strategy, MC3G generates realistic, actionable counterfactuals that closely mirror the original data, thereby overcoming key limitations of existing methods.

## 2.5. FOLD-SE

FOLD-SE, developed by Wang and Gupta (2024), is a rule-based machine learning algorithm that learns a compact stratified logic program to approximate a dataset, providing transparent and scalable decision-making. It is a part of the FOLD family of algorithms [Shakerin et al. (2017), Wang and Gupta (2022)] In MC3G, FOLD-SE serves as an explainable surrogate for any black-box classifier, enabling counterfactual generation without direct

access to the internals of the model while capturing causal dependencies among features. This integration unites model-agnosticism, explainability, and causal compliance.

## 3. Overview

### 3.1. The Problem

In high-stakes domains (e.g., loan approvals), black-box machine learning models might return negative outcomes without providing a guidance on how to obtain positive outcomes. Given the current feature vector $i$ for which the negative outcome holds, and a set of causally feasible positive outcomes $G$, the task is to find the minimal, causally consistent set of feature interventions that moves the individual from $i$ to some $g \in G$.

### 3.2. MC3G Approach

MC3G defines two distinct states: **Pre-intervention state** $i$: The current scenario where the model returns a negative outcome, and **Post-intervention state** $g \in G$: A feasible scenario where the model returns a positive outcome. Here $G$ is a set of all possible scenarios while $g$ is one such scenario that we wish to reach. MC3G traverses the individual from the **pre-intervention state** $i$ to the **post-intervention state** $g \in G$ (counterfactual).

MC3G first approximates the black-box model $f : X \to Y$ with a rule-based surrogate model $r : X \to Y$. This surrogate model, learned using a RBML algorithm (FOLD-SE), provides an interpretable representation of the decision logic of $f$, enabling transparent and causal-aware counterfactual reasoning. However just using these rules as an explanation is insufficient as in many scenarios revealing the underlying logic of the black box algorithm is undesired. Hence, the idea is to use counterfactuals that provide explanations by highlighting the minimal changes to the original instance $i$ (which obtained an undesired outcome) that are needed to obtain the desired outcome.

MC3G follows a three-step process: 1) Black-Box Model Approximation: The black-box model $f$ is approximated using a RBML algorithm (FOLD-SE), generating an explainable surrogate model $r$ that mimics the black-box model $f$'s decision-making. 2) Causal-Aware Counterfactual Search: Using ASP-based reasoning, the surrogate model $r$ and the original feature vector $i$ (negative outcome), MC3G identifies causally feasible changes that transition from $i$ to $g \in G$. 3) Optimized Cost Computation: Unlike prior methods, MC3G distinguishes between direct changes and automatic causal effects, ensuring cost is assigned only to user-initiated (direct) changes.

In ASP terms, the problem is formulated as: 1) Given a **pre-intervention state** $i$ where the query— `?- reject_loan(i).`—succeeds, 2) Compute causally consistent changes to transition to a **post-intervention state** $g \in G$ where the negation of the original query— `?- not reject_loan(g).`—succeeds. 3) Compute the counterfactual state $g \in G$ with minimal cost. To implement this, MC3G employs s(CASP), a query-driven ASP system, ensuring that counterfactual conditions hold while adhering to causal constraints. For Example, consider a loan application scenario, where an individual's approval depends on the following factors: 1) Debt Status: {no_debt, $\leq 10,000$, $> 10,000$}, 2) Bank Balance: {0, ..., 1000,000}, and 3) Credit Score: {300, ..., 850}. John {$Debt : > 10,000, Bank\ Balance : 40,000, Credit\ Score : 599$} applies for a loan and is denied.

**Step 1- Black-Box Approximation**: A black-box model $f$ predicts loan decisions based on the above features. MC3G approximates $f$ with a Rule-Based Machine Learning (RBML) model $r$ using FOLD-SE. This surrogate model learns interpretable decision rules,

enabling counterfactual generation without direct access to $f$. From $r$, we learn that the bank denies loans to applicants with: **Bank balance**: $< 60,000$ and **Credit score**: $< 600$.

**Step 2- Identifying Counterfactuals**: John $\{Debt : > 10,000, Bank\ Balance : 40,000, Credit\ Score : 599\}$ meets the conditions of $r$. Hence, he is denied the loan, i.e., the query —?- reject_loan(john).—is $TRUE$. John is in the Pre-Intervention state $i$.

**Incorrect Counterfactual (Ignoring Causality)**: A naive counterfactual assumes feature independence and suggests: 1) Increase **Bank balance** to $60,000$, and 2) Increase **Credit score** to 620. This is unrealistic, as the **Credit score** cannot be arbitrarily changed. It fails to recognize that the **Credit score**'s improvements depend on reducing **Debt**.

**MC3G's Causal-Aware Counterfactual**: MC3G correctly models causal dependencies, recognizing that: Clearing **Debt** leads to a higher **Credit Scores**. Thus, a realistic counterfactual solution is: John $\{Debt : no\ debt, Bank\ Balance : 60,000, Credit\ Score : 620\}$. The required interventions are: 1) Increase **Bank balance** to $60,000$, and 2) Clear **Debt**, which automatically increases the credit score to 620. By respecting causal constraints, MC3G ensures that only feasible interventions are suggested, avoiding unrealistic modifications that do not respect causal dependencies.

**Step 3- Finding Minimal Counterfactual**: Assign *zero* weight to causal changes so that only direct, user-initiated changes contribute to the overall cost. The counterfactual with the lowest resulting cost is therefore the minimal causally compliant solution.

### 3.3. Cost Computation in MC3G

**Standard Cost Computation** used by existing methods (C3G, MACE) assigns a cost to every feature change, even those that occur automatically from causal effects. It weights direct interventions (user initiated) and causal effects equally, unfairly penalizing causally compliant solutions by making them costly. **MC3G's Refined Cost Computation** distinguishes between direct interventions and causal effects. It excludes automatic changes due to causal effects from the cost computation. Hence, it does not artificially inflate causally compliant solutions

For example, in John's case:, **Standard Cost Computation** treats an "increase in **Credit Score**, **Bank Balance**" and "clearing **Debt**" as three interventions, inflating the cost. In contrast, **MC3G's Refined Cost Computation** sees that clearing **Debt** automatically improves **Credit Score**, so only the direct interventions (clear **Debt**; increase **Bank Balance**) contribute to the cost. By refining the cost computation, MC3G ensures that the realistic counterfactuals with the lowest cost are prioritized, making recourse strategies both actionable and fair.

## 4. Methodology

Unlike standard counterfactual methods that assume feature independence, MC3G models causal dependencies, yielding realistic counterfactual recommendations. This section outlines the key components of MC3G's counterfactual-generation framework.

### 4.1. Definitions

#### 4.1.1. STATE SPACE (S)

$S$ represents all combinations of feature values. For domains $D_1, ..., D_n$ of the features $F_1, ..., F_n$, $S$ is a set of possible states $s$, where each state is defined as a tuple of feature values $V_1, ..., V_n$. $s \in S$ where $S = \{(V_1, V_2, ..., V_n) \mid V_i \in D_i,\ for\ each\ i\ in\ 1, ..., n\}$. *E.g., an individual John:* $s = (> \$10,000, \$40,000, 599\ points)$, *where* $s \in S$.

### 4.1.2. Causally Consistent State Space ($S_C$)

$S_C$ is a subset of $S$ where all causal rules are satisfied. $C$ represents a set of causal rules over the features within a state space $S$. Then, $\theta_C : P(S) \to P(S)$ (where $P(S)$ is the power set of S) is a function that defines the subset of a given state sub-space $S' \subseteq S$ that satisfy all causal rules in C. $\theta_C(S') = \{s \in S' \mid s \ satisfies \ all \ causal \ rules \ in \ C\}$ and $S_C = \theta_C(S)$. E.g., causal rules state that if *debt* is 0, the credit score should be above 599, then instance $s_1 = (no \ debt, 40000, 620 \ points)$ is causally consistent, and instance $s_2 = (no \ debt, 40000, 400 \ points)$ is causally inconsistent.

### 4.1.3. Decision Consistent State Space ($S_Q$)

$S_Q$ is a subset of $S_C$ where all decision rules are satisfied. $Q$ represents a set of rules that compute some external decision for a given state. $\theta_Q : P(S) \to P(S)$ is a function that defines the subset of the causally consistent state space $S' \subseteq S_C$ that is also consistent with decision rules in $Q$. $\theta_Q(S') = \{s \in S' \mid s \ satisfies \ any \ decision \ rule \ in \ Q\}$. Given $S_C$ and $\theta_Q$, we define the decision consistent state space: $S_Q = \theta_Q(S_C) = \theta_Q(\theta_C(S))$. E.g., John whose loan has been rejected: $s = (no \ debt, \$40000, 620 \ points)$, where $s \in S_Q$.

### 4.1.4. Counterfactual Generation (CFG) Problem

A counterfactual generation (CFG) problem is a 3-tuple $(S_C, S_Q, I)$ where $S_C$ is the causally consistent state space, $S_Q$ is the decision consistent state space, $I \in S_C$ is the initial state.

### 4.1.5. Goal Set $G$

The goal set $G$ is the set of desired outcomes that do not satisfy the decision rules $Q$. For the Counterfactual Generation (CFG) problem $(S_C, S_Q, I)$, $G \subseteq S_C$, we have $G = s \in S_C \mid s \notin S_Q$. $G$ includes all states in $S_C$ that do not satisfy $S_Q$. For **example**, $g = (no \ debt, 60000, 620 \ points)$ where $g \in G$.

### 4.1.6. Finding a Counterfactual Solution

A solution to the problem $(S_C, S_Q, I)$ with Goal set $G$ is any state $g \in G$. This means a valid counterfactual must satisfy two conditions: 1) It respects causal constraints ($g \in S_C$), and 2) It achieves the desired outcome ($g \notin S_Q$. Example: A valid counterfactual solution is John $Debt : no \ debt, Bank \ Balance : 60,000, Credit \ Score : 620$. Here, John clears his debt, which naturally increases his credit score and raises his bank balance. This ensures that he is qualified to have his loan approved.

## 4.2. Algorithm to Obtain the Counterfactual

### 4.2.1. Algorithm 1: **MC3G**

*MC3G* combines three algorithms in sequence to find minimal-cost counterfactuals that overturn an undesired decision. First, *extract_logic* (found in the supplement) obtains decision rules from the original model—directly if the model is already rule-based, or by training a rule-based surrogate on the model's predictions. Next, for each candidate state in the dataset, *is_counterfactual* checks whether it satisfies causal constraints and avoids the undesired decision, while also zeroing out the weights of any features changed automatically by those causal dependencies. This means causally compliant changes contribute **nothing** to the cost. Finally, *compute_weighted_Lp* measures the overall distance from the initial state to each valid counterfactual, taking into account only direct user-initiated changes (since features altered by causality have zero weight). The method then selects the counterfactual with the lowest total cost as the optimal solution.

**Algorithm 1:** MC3G: Generating Counterfactuals & Selecting Minimum Cost
**Input:** Original Model $M$, Data $H$, RBML Algorithm $R$, Set of Causal Rules $C$, Initial
State $s_0$, Candidate States $S$, Feature Weights $W$, Norm Parameter $p \in \{0, 1, 2\}$
**Output:** Optimal Counterfactual State $s^*$, Minimum Cost *bestCost*

```
// 1.  Extract Decision Rules from the Model
```
$Q \leftarrow \textbf{extract\_logic}(M,\ H,\ R)$

$bestCost \leftarrow \infty$ `// Initialize best cost to a large value`
$s^* \leftarrow \textbf{NULL}$ `// Optimal counterfactual state not found yet`

**foreach** *state* $s \in S$ **do**
    `// 2.  Check if s is a counterfactual and adjust weights`
    $(isValid,\ adjWeights) \leftarrow \textbf{is\_counterfactual}(s,\ C,\ Q,\ W)$
    **if** $isValid = TRUE$ **then**
        `// 3.  Compute cost from initial state s0 to s`
        $cost \leftarrow \textbf{compute\_weighted\_Lp}(s_0,\ s,\ adjWeights,\ p)$
        **if** $cost < bestCost$ **then**
            $bestCost \leftarrow cost \quad s^* \leftarrow s$
        **end**
    **end**
**end**
**return** $(s^*,\ bestCost)$ `// Return the state with minimal cost and its cost`

**Algorithm 2: extract_logic**: Extract the underlying logic of the classification model
**Input:** Original Classification model $M$, Data $H$, *RBML* Algorithm $R$
**if** $M$ *is rule-based* **then**
    $Q \leftarrow M$ `// Decision Rules are the rules of M`
**end**
**else if** $M$ *is statistical* **then**
    $V \leftarrow \textbf{predict}(M(H))$ `// For input data H, predict the labels as V`
    $R \leftarrow \textbf{train}(R(H,V))$ `// Use the H and V to train R`
    $Q \leftarrow R$ `// Decision Rules are the rules of R`
**end**
**return** $Q$

### 4.2.2. ALGORITHM 2: **extract_logic**

We first describe the algorithm for extracting the underlying logic in the form of rules for the classification model that provides the undesired outcome. By using this extracted logic or rules, we can generate a path to the counterfactual solution $g$. The function '***extract_logic***' extracts the underlying logic of the classification model used for decision-making. Our *MC3G* framework applies specifically to tabular data, so any classifier handling tabular data can be used. Algorithm 2 provides the pseudocode for '***extract_logic***', which takes the original classification model $M$, input data $H$, and a *RBML* algorithm $R$ as inputs and returns $Q$, the underlying logic of the classification model. $Q$ represents the decision rules responsible for generating the undesired outcome. The algorithm first checks if the classification model $M$ is rule-based. If yes, we set $Q = M$ and return Q. Otherwise, the

corresponding labels for the input data are predicted using $M$. These predicted labels, along with the input data $H$, are then used to train the $RBML$ algorithm $R$. The trained $RBML$ algorithm represents the underlying logic of $M$, and we set $Q = R$, returning $Q$ as the extracted decision rules.

### 4.2.3. Algorithm 3: **is_counterfactual**

> **Algorithm 3: is_counterfactual**: Checks if a state $s$ is a valid counterfactual while adjusting feature weights

**Input:** State $s \in S$, Set of Causal Rules $C$, Set of Decision Rules $Q$, Feature Weights $W$
**Output:** Boolean indicating if $s$ is a counterfactual, Updated Feature Weights $W'$
adjusted_weights $\leftarrow W$ // Initialize weights for feature changes
**foreach** *feature* $F_k \in s$ **do**
     **if** $F_k$ ***was altered due to a causal dependency in*** $C$ **then**
         adjusted_weights$[F_k]$ $\leftarrow$ 0 // Set weight to 0 if change was caused by a
            causal dependency
     **end**
**end**
**if** ***is_causally_consistent(s, C) AND not is_decision_compliant(s, Q)*** **then**
     **return** ($TRUE$, adjusted_weights) // $s$ is a valid counterfactual
**end**
**else**
     **return** ($FALSE$, adjusted_weights) // $s$ is not a valid counterfactual
**end**

This algorithm determines whether a given state $s$ qualifies as a valid counterfactual by checking two conditions: (1) is it causally consistent (it satisfies all causal rules), and (2) does it **not** satisfy the decision rules. Additionally, it sets the weights of any features whose changes occur automatically due to causal dependencies to 0, preventing these "free" adjustments from inflating the overall cost. If both conditions are met, the algorithm returns $TRUE$ and the *adjusted feature weights*, indicating that $s$ is a valid counterfactual. Otherwise, it returns $FALSE$ and the *adjusted weights*.

### 4.2.4. Algorithm 4: **compute_weighted_Lp**

This algorithm calculates the overall cost of transforming one state $s$ into another state $s' = g$ under three possible distance metrics—$L_0$, $L_1$, *or* $L_2$— while incorporating feature-specific weights. It loops over each feature feature $F_k$, checks if the user-specified norm parameter $p$ is 0, 1, or 2, then computes the contribution of changing $F_k$ accordingly. Specifically, $L_0$ counts how many features differ (adding the corresponding weight if a feature changed), $L_1$ sums the weighted absolute differences, and $L_2$ sums the weighted squared differences. Crucially, if any feature's weight is zero (for example, because it was altered automatically due to causal dependencies), it contributes nothing to the distance, ensuring only direct user-initiated changes inflate the overall distance/cost. By summing these per-feature contributions, the algorithm outputs a single numeric cost reflecting the magnitude of the change required to transform $s$ into $s' = g$.

**Algorithm 4: compute_weighted_Lp**: Computes weighted $L_0$, $L_1$, or $L_2$ cost between two states

**Input:** States $s, s' \in S$, Feature Weights $W$, Norm Parameter $p \in \{0, 1, 2\}$

**Output:** Overall Cost $cost$

$cost \leftarrow 0$ // Initialize total cost

**foreach** *feature $F_k$ in s* **do**

   **if** $p = 0$ **then**

      // L0 norm counts number of changed features

      **if** $s'[F_k] \neq s[F_k]$ **then**

         | $cost \leftarrow cost + W[F_k]$

   **end**

   **else if** $p = 1$ **then**

      // L1 norm sums absolute differences

      $\Delta \leftarrow |s'[F_k] - s[F_k]| \quad cost \leftarrow cost + W[F_k] \times \Delta$

   **end**

   **else if** $p = 2$ **then**

      // L2 norm sums squared differences

      $\Delta \leftarrow (s'[F_k] - s[F_k]) \quad cost \leftarrow cost + W[F_k] \times (\Delta)^2$

   **end**

**end**

**return** $cost$

## 5. Experiment

### 5.1. Generate Causally Constrained Counterfactuals

Table 1: Performance of MC3G against counterfactual based methods

| Dataset | Model | Causally Compliant | Causal Consistency (%) |
|---------|-------|--------------------|-----------------------|
| **Adult** | Borderline-CF | FALSE | 30 |
| | DiCE | Indirectly | 80 |
| | MACE | FALSE | 80 |
| | C3G | **TRUE** | **100** |
| | **MC3G** | **TRUE** | **100** |
| **Statlog** | Borderline-CF | FALSE | 80 |
| | DiCE | Indirectly | 80 |
| | MACE | FALSE | 20 |
| | C3G | **TRUE** | **100** |
| | **MC3G** | **TRUE** | **100** |
| **Car** | Borderline-CF | FALSE | N/A |
| | DiCE | Indirectly | N/A |
| | MACE | FALSE | N/A |
| | C3G | **TRUE** | **N/A** |
| | **MC3G** | **TRUE** | **N/A** |

To see the kinds of counterfactuals produced, we use datasets that have well defined causal dependencies known to us: the Adult dataset by Becker and Kohavi (1996) and the Statlog (German Credit) dataset by Hofmann (1994). For comparison, we use the Car Evaluation dataset for which no causal dependencies are used. We compare our MC3G method against Borderline Counterfactuals, DiCE, MACE and C3G and check the type of counterfactuals produced. From Table 1, we see that MC3G like C3G produces causally compliant counterfactuals 100% of the time. The code for MC3G is provided by Dasgupta (2025).

## 5.2. Comparison of Counterfactual Proximity

Table 2 demonstrates that MC3G consistently produces closer counterfactuals than C3G across all metrics—nearest, furthest, and average distances—regardless of norm—L1 or L2 norm—used. This is because MC3G correctly accounts for causal dependencies, treating causally induced changes as cost-free, whereas C3G incorrectly assigns a cost to all feature modifications, including those that occur naturally. For both the Adult and German datasets, MC3G counterfactuals exhibit lower nearest (K=1), furthest (K=20), and average distances (Avg.) than C3G. This highlights that MC3G identifies more efficient intervention strategies, ensuring that users receive recourse recommendations requiring minimal effort while remaining causally compliant. The reduced distance across all metrics confirms that MC3G outperforms C3G in generating counterfactuals that are not only feasible but also require fewer modifications to achieve the desired outcome.

Table 2: Comparison of Nearest and Furthest Counterfactuals for C3G and MC3G

| Dataset | Model | K = 20 | | | | | | | | |
| | | Metric Used to Sort the Closest | | | | | | | | |
| | | L1 | | | L2 | | | L0 | | |
| | | K=1 | K=20 | Avg. | K=1 | K=20 | Avg. | K=1 | K=20 | Avg. |
| Adult | C3G | 1.012 | 1.470 | 1.296 | 1.012 | 1.221 | 1.113 | 1 | 1 | 1 |
| | MC3G | 0.701 | 1.031 | 0.907 | 0.701 | 0.903 | 0.829 | 1 | 1 | 1 |
| German | C3G | 3.334 | 3.341 | 3.337 | 1.764 | 1.764 | 1.764 | 4 | 4 | 4 |
| | MC3G | 2.334 | 2.341 | 2.337 | 1.453 | 1.453 | 1.453 | 3 | 3 | 3 |
| Cars | C3G | 1 | 3 | 2.3 | 1 | 1.732 | 1.499 | 1 | 3 | 2.3 |
| | MC3G | 1 | 3 | 2.3 | 1 | 1.732 | 1.499 | 1 | 3 | 2.3 |

## 6. Conclusion and Future Work

In this paper, we introduced MC3G, a novel, model-agnostic framework for generating causally consistent counterfactual explanations. Unlike previous methods—including C3G and other approaches that either neglect causal dependencies or fail to distinguish between user-initiated and automatic changes—MC3G leverages Answer Set Programming to ensure that every counterfactual adheres to real-world causal relationships while computing intervention costs accurately. By zeroing out the cost contributions from automatically induced feature changes, MC3G delivers more realistic, actionable, and cost-efficient recourse for individuals affected by adverse decisions. While MC3G currently incurs a higher computational cost and is limited to tabular data, future work will focus on optimizing the search space and extending the framework to handle non-tabular data, such as images. Overall, MC3G represents a significant advancement in counterfactual explanation methods, enhancing both interpretability and practical utility in diverse decision-making contexts.

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
