# OpenReview forum: "MC3G: Model Agnostic Causally Constrained Counterfactual Generation"
_nesyconf.org/NeSy/2025/Conference — NeSy 2025 Poster_

### Official Review · Reviewer_sKMc · 2025-04-02
**Review for MC3G**

**Rating:** 6
**Confidence:** 4

**Review:**

Summary: The authors present MC3G, a framework designed to generate counterfactual explanations for machine learning models, emphasizing transparency and actionable recourse in high-stakes decision-making contexts. MC3G is model-agnostic, approximating any black-box model with an interpretable rule-based surrogate. It also refines cost computation to consider only user-initiated changes, thereby providing a realistic assessment of the effort required to achieve a favorable outcome.

Strengths:
It prevents exposing model internals while provides interpretable and actionable recommnedation. interesting for stakeholders in high-stakes environments such as finance, law, and hiring.

Weaknesses and Questions:

Relation to NeSy: Seems the methodology is predominately symbolic-based and lacks of neural parts. Clarification on how this approach aligns with NeSy systems would be beneficial.

Surrogate Model Fidelity: The effectiveness of MC3G hinges on the accuracy of the rule-based surrogate model in approximating the original black-box model. Further details on the fidelity of this approximation and its impact on counterfactual generation would be beneficial.​ It would be interesting to show some cases that ruled-based model not align with the original one.

Scalability and Performance: Insights into MC3G's scalability, particularly when dealing with high-dimensional datasets or complex causal structures, are necessary to understand its practical applicability in large-scale scenarios.​

Related works like CoGS and BayCon might be interesting to the authors.

**Anonymity:**

Remain anonymous

---

### Official Review · Reviewer_bSc1 · 2025-04-07

**Rating:** 6
**Confidence:** 4

**Review:**

The paper presents MC3G (Model-Agnostic Causally Constrained Counterfactual Generation), a novel framework for generating counterfactual explanations. MC3G aims to overcome limitations in existing methods by being model-agnostic, meaning it can be applied to any black-box machine learning model. It achieves this by first approximating the black-box model with an explainable rule-based model using FOLD-SE. Furthermore, MC3G explicitly models causal dependencies to ensure the generated counterfactuals are realistic. A key feature of the framework is its refined cost computation, which distinguishes between user-initiated interventions and automatic feature adjustments due to causal relationships, assigning cost only to the former.

Conversely, MC3G seems to have a high computational cost. However, this aspect is not explored in detail within the main body of the paper, and a comparison of computational complexity with other methods is lacking. Understanding the scalability of MC3G would be important for practical applications.

Furthermore, while FOLD-SE achieves high fidelity in approximating the black-box models, Table 1 shows that there is still a slight performance trade-off (in terms of accuracy, precision, recall, and F1-score) compared to the original DNN, GBC, and RF models. While the loss is mostly negligible, the potential impact of this approximation on the quality and reliability of the generated counterfactuals could be discussed further.

MC3G presents an advancement in the field of counterfactual explanation generation by introducing a model-agnostic framework that explicitly incorporates causal dependencies and refines cost computation. The approach leveraging FOLD-SE and ASP is a key strength, enabling the generation of more realistic and actionable explanations for black-box models. The empirical results demonstrate the framework's ability to produce closer counterfactuals compared to existing methods like C3G.

To further strengthen the paper, the authors could focus on (i) Providing a more thorough analysis of the computational cost and potential scalability challenges of MC3G and (ii) Expanding the discussion on the implications of the fidelity trade-off in the black-box approximation.

**Anonymity:**

Remain anonymous

---

### Official Review · Reviewer_8ZdH · 2025-04-07
**A combined method to generate model agnostic, causally constrained counterfactuals with improved cost behaviour**

**Rating:** 6
**Confidence:** 3

**Review:**

This paper proposes a method called MC3G, which combines two existing methods with an improved cost modelling to allow the generation of realistic and useful counterfactual examples while hiding model internals. The use of FOLD-SE for replacing a classifier with a rule based model means that direct access to the original model is not needed, and information about the original model. Answer set programming ensures that causal constraints are applied. The exclusion of cost for changes due to causal constraints, rather than input change, mean that the the counterfactuals produced by MC3G.

The main strength of the paper is in solving the practical problem of keeping model information secret while providing counterfactuals that follow domain logic and use a realistic cost function to improve the quality of counterfactuals.
The weaknesses are in the lack of novelty, where the method is mainly a recombination and the evaluation uses only a few not very large datasets. There are a number of questions that remain to be addressed: to what extent can model information be extracted from a black-box model and to what extent can it be prevented by using a surrogate? how realistic is the assumption that an input change that is caused by another change does not entails cost? (e.g. if income is higher, my other taxes caused in the sense that they may be unavoidable, but they are still a cost). How can the quality of the counterfactuals be measured in a way that is independent of the cost function, e.g. by human experiments?

**Anonymity:**

Remain anonymous